# A Systematic Review of the Efficacy of Microfocused Ultrasound for Facial Skin Tightening

**DOI:** 10.3390/ijerph20021522

**Published:** 2023-01-13

**Authors:** Mark Contini, Marijke H. J. Hollander, Arjan Vissink, Rutger H. Schepers, Johan Jansma, Jurjen Schortinghuis

**Affiliations:** 1JC Kliniek, Boermarkeweg 44C, 7824 AA Emmen, The Netherlands; 2Department of Oral and Maxillofacial Surgery, University Medical Center Groningen (UMCG), 9700 RB Groningen, The Netherlands

**Keywords:** microfocused ultrasound, HIFU, skin, wrinkle, laxity, rejuvenation, cosmetic

## Abstract

Objective: to systematically review the efficacy of microfocused ultrasound (MFU) for facial skin tightening. Methods: A systematic search was performed (Pubmed, Embase) to assess the efficacy of single MFU treatments for facial skin tightening. Eligible studies included randomised controlled trials, controlled trials, cohort studies and case series (n ≥ 10). Objective and subjective outcomes were assessed. Results: A total of 693 studies were identified of which 16 studies were eligible. All the studies involved female patients. MFU is capable of tightening the skin, as observed in studies measuring the results of brow lifts (0.47–1.7 mm) and submental lifts (measured as a 26–45 mm^2^ reduction in the submental area on lateral photographs). Data from the Global Aesthetic Improvement Scale (GAIS) were pooled, and the day 90 pooled subjective investigator reported scores (IGAIS) (n = 337) showed that 92% of the patients demonstrated an improvement in skin tightening and/or in wrinkle reduction which continued up to one year. Longer-term follow-up data are not available. The patient-reported pooled scores (SGAIS) (n = 81) showed that the skin improvements were mild and continued to increase from 42% (90 days) to 53% (360 days) post-treatment. The MFU treatment was moderately painful and caused transient erythema with or without oedema. Other adverse effects were rare (2%), including dysesthesia (numbness or hypersensitivity), bruising and stinging, mandibular burns, striations and contact dermatitis. Various device settings, treatment protocols and energies were applied. Excessive skin laxity and a BMI > 30 were posed as relative contraindications for MFU treatment because positive results declined with an increase in laxity and BMI. Conclusions: MFU treatment is effective in tightening female patients’ mildly to moderately lax facial skin. Future studies should focus on objective treatment outcomes, optimising treatment regimens and male patients.

## 1. Introduction

The aim of microfocused ultrasound (MFU), a noninvasive treatment method, is to tighten the skin. It is an energy-based modality that induces tissue damage followed by tissue necrosis, whereby the energy of the ultrasound waves is converted into heat and cavitation [1]. The target areas are the subdermal connective tissues, such as the superficial muscular aponeurotic system (SMAS) layer and (deep) dermal layers. Multiple, small thermal injury zones (TIZ) of about 1 mm^3^ in size are created at predetermined depths, the aim being not to damage the surrounding tissues [2,3]. TIZ result in immediate collagen contraction and denaturation that induce neocollagenesis and neoelastogenesis for more than one year [4,5,6]. Together, these processes are thought to contribute to the tightening of the skin for rejuvenation purposes. Although several studies in the literature measured the effect of MFU objectively and subjectively on skin tightening, more rigorous data are missing. Therefore, the literature on the efficacy of MFU for facial skin tightening was reviewed systematically.

## 2. Methods

This systematic review was prepared according to the Preferred Reporting Items for Systematic and Meta-Analysis (PRISMA) guidelines [7]. A search of the international prospective register of systematic reviews (Prospero) did not unearth any systematic reviews regarding this topic on the date of the search (19-08-2021). The protocol ID is 179974.

### 2.1. Search Strategy

The Pubmed and Embase databases were searched with these keywords: “high intensity focused ultrasound”, “focused ultrasound” and the abbreviations “HIFU” (High-Intensity Focused Ultrasound), “IFUS” (Intense Focused Ultrasound), “MFU” as well as “skin”, “tightening”, “rejuvenation”, “laxity”, “cosmetic”, “rhytids”, or, “wrinkle”, (Appendix A). The search was conducted to include the scope of ultrasound AND skin to select publications that deal with ultrasound and skin; to select publications that deal with skin rejuvenation more specifically, different words addressing skin rejuvenation were selected so that at least one of them would be present (OR). The reference lists of the included studies were screened to find any publications missing from the search.

### 2.2. Eligibility Criteria

Eligible studies were randomised controlled trials, controlled trials, cohort studies and case-control studies with ≥10 participants. The length of the follow-up had to be ≥3 months. No language restrictions were applied (Appendix B). The aim of the studies was an investigation of the efficacy of MFU on skin tightening, skin laxity and wrinkles after a single session of treatment. If the same treatment protocol was studied on different areas of the body but the face was evaluated separately, the study was also considered eligible for inclusion. When treatment protocols were combined in the same area, the study was excluded. The primary outcome measurement was skin improvement (measured as grades in improvement, skin tightening, skin laxity and/or wrinkles). The secondary outcome was adverse effects.

### 2.3. Study Selection

MC performed the search strategy and removed all duplicates. The selection of studies for inclusion was carried out by two observers (MC, MH) by first checking if any of the eligibility criteria appeared in the title or abstract. Then, the full texts of the selected articles were evaluated further. If there was uncertainty because the abstract or title did not provide sufficient information as to whether the selection criteria had been met, the full text articles were also evaluated. An insight into the selection decisions is depicted in a flow chart (Appendix C).

### 2.4. Data Extraction

The data were extracted from the included papers by one observer (MC) and noted on standardised extraction forms. The second observer (MH) carried out an independent check of the data extraction forms to ensure that all was complete and correct.

### 2.5. Data Synthesis

All the methods used to assess the effect of MFU with the selected studies were evaluated (for example brow lifting). The studies’ results were pooled when possible. Descriptive statistics were used.

## 3. Results

### 3.1. Study Selection

A total of 693 studies were identified for inclusion and screened after removing duplicates.

There was a disagreement about including 23 studies based on the abstract. These studies were discussed in a consensus meeting and, if a disagreement persisted, a third observer (JS) was available to give a binding verdict. After the abstract screening, a total of 43 full texts were assessed. The excluded studies did not have any of the eligibility criteria that appeared in the title or abstract. The initial search was in June 2019, and the last update was in June 2021. In this period, another five studies were included, leading eventually to 16 studies being eligible for analysis (see the flowchart in Appendix C for the selection process).

### 3.2. Interobserver Agreement

After assessing the titles and abstracts, the agreement between the two observers (MC and MH) was 96%. A further discussion resulted in a consensus of 100% with a Cohen’s kappa of 1.0.

### 3.3. Study Characteristics

The included study characteristics are displayed in Table 1 and Table 2. All of the 16 included studies were eligible for qualitative analysis and six of them also for quantitative analysis (Table 3). Six studies evaluated the face as a whole whereas 12 studies analysed parts of the face (also) separately.

### 3.4. Treatment Effects

#### 3.4.1. Objective Measurements (Table 3)

The objective measurements applied in the studies were diverse. Skin tightening after MFU was expressed as the number of millimetres of vertical displacement of the brows after treating the forehead and periorbital region [12,14,17,18,22], the number of square millimetres of submental lift after a full-face treatment [10,12,14] and the number of millimetres of oblique displacement of the marionette lines [18]. These measurements were done on 2D light photographs taken at baseline and different follow-up periods, typically 90 days (3 months), 180 days (6 months) and 360 days (1 year).

The absolute brow lifts varied from 0.47 mm (*p* < 0.02) [14] to 1.25 mm (*p* < 0.00) [22] and 1.7 mm at 3 months [17] post-treatment. Only one study gave 6-month results (1.22 mm) [22]. Sasaki et al. [18] depicted the distance between the baseline and brow edges as a percentage and reported a brow lift of 5.6–7.2% 6 months after applying the MFU treatment lines in a superolateral direction and a brow lift of 1–3.7% after applying the treatment lines in a horizontal direction. Werschler and Werschler [12] reported a percentage of patients with a brow lift of >0.5 mm. At 3 months, 35% of the subjects had a brow lift of more than 0.5 mm, and this had increased to 45% after 6 months but then decreased to 39% by the 1-year follow-up.

When treating the submental area, skin tightening may decrease the laxity of the submental tissues leading to a ‘lift’ of this area. In lateral photographs, the submental area appears smaller. Oni et al. [10] measured an average reduction of 45 mm^2^ at 3 months. A lift could be seen in 56 of the 78 treated Caucasian patients. A double pass technique was used to treat the skin at two different depths (3.0 mm and 4.0 mm). Lu et al. [14] used the same method and found a reduction of 26 mm^2^ after 3 months and 14 mm^2^ after 6 months in Asian patients. Sasaki et al. [18] measured the displacement of the marionette lines 3 months after an MFU treatment. The lift varied between the treatment regimens from 2.4 to 3.8% along a line from the inferior tragal notch to the midpoint of the marionette line.

#### 3.4.2. Subjective Measurements (Table 4 and Table 5)

Different scoring systems were used to assess the subjective treatment results. The Investigator Global Aesthetic Improvement Scale (IGAIS, Table 4) rates the improvement or worsening in skin laxity on a 4-point scale [9,13,18,23] or a 5-point scale [12,14,19,24]. Both the 4- and 5-point IGAISs have the same four categories ranging from ‘no change’ to ‘significant improvement’, but the 5-point scale includes the item ‘worsened’.

**Table 4 ijerph-20-01522-t004:** Investigator Global Aesthetic Improvement Scale (IGAIS). IGIAS 4-point scale, IGIAS 5-point scale and pooled scores are presented that combine the 4 and 5 point scale.

IGAIS 4-Point Scale 0 = No Change, 1 = Mild Improvement, 2 = Moderate Improvement, 3 = Significant Improvement			IGAIS 5-Point Scale 1 = Very Much Improved, 2 = Marked Improvement, 3 = Improved, 4 = No Change, 5 = Worse		
Author	Outcomes (%) Day 90	Outcomes (%) Day 180	Author	Outcomes (%) Day 90	Outcomes (%) Day 180
Sasaki et al. [18] Study group 1 N = 107	0: – – 1: 47 2: 53 3: – –	0: – – 1: 31 2: 69 3: – –	Lu et al. [9] N= Day 90 21 Day 180 22	1: – – 2: 16 3: 68 4: 16 5: – –	1: – – 2: 16 3: 72 4: 12 5: – –
Sasaki et al. [18] Study group 2 N = 55	A 0: – – 1: 34 2: 48 3: 18	0: – – 1: 10.4 2: 63.4 3: 26.2	Fabi et al. [17] N= Day 90 16 Day 180 45	1: 0 2: 44 3: 37 4: 19 5: – –	1: 13 2: 18 3: 47 4: 22 5: – –
Ko et al. [9] Cheeks only N = 32	0: 9 1: 91 2: – – 3: – –		Werschler et al. [8] N= Day 90 20 Day 180 19	1: 5 2: 55 3: 40 4: – – 5: – –	1: 5 2: 53 3: 42 4: – – 5: – –
Lee et al. [13] Lower face N = 12	0: 20 1: 20 2: 40 3: 20		Yalici-Armagan et al. [18] Lower face N = 24	1: – – 2: 4 3: 17 4: 63 5.17	
Shome et al. [23] Mid and lower face N = 50	0: – – 1: 52 2: 24 3: 24				
**Pooled Investigator Reported Scores** (Converted scores) IGAIS 4- and 5-point score, all facial areas: score 0 = no change, 1 = mild improvement, 2 = moderate improvement, 3 = significant improvement, 4 = worsening. Data presented as percentage and number of cases.					
	Day 90 (n = 337)	Day 180 (n = 249)			
No change (0) Mild improvement (1) Moderate improvement (2) Significant improvement (3) Worsening (4)	7% (n = 25) 47% (n = 159) 36% (n = 122) 8% (n = 26) 1% (n = 4)	5% (n = 13) 34% (n = 84) 52% (n = 130) 8% (n = 21) 0%			

In the pooled IGAIS (n = 337), a total of 92% of the patients demonstrated an improvement at day 90; the ‘mild improvement’ category was greater (47%) than the ‘moderate improvement’ category (36%) (Table 4). A shift was observed after the 180-day evaluation: Although the majority of patients still had ‘mild’ and ‘moderate’ improvement scores, the moderate category had increased (52%). ‘No change’ or ‘worsening’ was only present in a minority of patients (<7%). The patients could also self-evaluate the effect of the MFU treatment using the 4-point Subject Global Aesthetic Improvement Scale (SGAIS) (Table 5). The pooled SGAIS (n = 81) was predominantly ‘mild improvement’, which increased from 42 to 53% in the period 90 to 360 days post-treatment. The ‘moderate improvement’ category also increased from 13 to 32% between the different time intervals. Worsening did not occur, whereas the ‘no change’ category decreased with time from 25% at 90 days to 5% at 360 days. Of note is that Fabi et al. [24] reported a significantly higher SGAIS score among patients with a BMI of ≤25 kg/m^2^. Oni et al. [10] reported less improvement in subjects with a BMI exceeding 30 kg/m^2^. Out of the 11 patients whose BMI exceeded 30 kg/m^2^, only three showed improvement.

**Table 5 ijerph-20-01522-t005:** Subject Global Aesthetic Improvement Scale (SGAIS) and pooled subjective scores.

SGAIS 5-Point Scale. 1 = Very Much Improved, 2 = Much Improved, 3 = Improved, 4 = No Change, 5 = Worse			
Author	Outcomes (%) Day 90	Outcomes (%) Day 180	Outcomes (%) Day 360
Lu et al. [14] N= Day 90 21 Day 180 22	1: 4 2: 12 3: 68 4: 16 5: – –	1: – – 2: – – 3: 80 4: 20 5: – –	
Fabi et al. [24] N= Day 90 16 Day 180 45	1: 2: 19 3: 56 4: 25 5 – –	1: 13 2: 9 3: 56 4: 22 5:– –	
Werschler et al. [12] N= Day 90 20 Day 180, 360 19	1: 25 2: 25 3: 40 4: 10 5: – –	1: 16 2: 37 3: 42 4: 5 5: – –	1: 11 2: 32 3: 53 4: 5 5: – –
Yalici-Armagan et al. [19] N = 24 Lower face	1: 5 2: 40 3: 10 4: 45 5: – –		
**Pooled patient-reported scores** (Converted) SGAIS 5-point scale: all facial areas: score 0 = no change, 1 = mild improvement, 2 = moderate improvement, 3 = significant improvement, 4 = worsening. Data presented as percentage and number of cases.			
	Day 90 (n = 81)	Day 180 (n = 86)	Day 360 (n = 19)
No change (0) Mild improvement (1) Moderate improvement (2) Significant improvement (3) Worsening (4)	25% (n = 20) 42% (n = 34) 25% (n = 20) 9% (n = 7) 0%	17% (n = 15) 59% (n = 51) 13% (n = 11) 10% (n = 9) 0%	5% (n = 1) 53% (n = 10) 32% (n = 6) 11% (n = 2) 0%

Park et al. [11] used a self-modified photographic scale evaluating seven facial areas in 20 patients. Each area was scored from 0 (no wrinkles) to 4 (severe). A HIFU treatment decreased the wrinkle score by 0.9 points at 3 and 6 months, with the most effect on the jawline and cheek area.

Saket et al. [16] evaluated the MFU effect with an ‘observation’ scoring system ranging from 10 (no efficacy on wrinkles and lifting effect) to 100 (maximum efficacy). After 3 months post-treatment, most of the 22 patients treated scored between 58 and 66 (physician rated) or 50–60 (patient-rated). 

Friedman et al. [20] treated lower face and neck laxity with ultrasound in 43 patients. Based on a global grading scale from 0 (exacerbation) to 5 (75–100% improvement), at three months, an improvement was seen in nine patients who had only slight sagging. The others showed no effect.

Araco et al. [21] used his self-developed Surgeon Assessment Scoring System scores and patient satisfaction questionnaire scores in 50 patients who had one MFU treatment. After 6 months, the surgeon’s score was 80, indicating a moderate effect on skin texture and a minimal-to-moderate face-lifting effect as visible in photographs. Patients indicated a ‘moderate’ to ‘good’ difference after treatment.

Other evaluation methods are the Fasil Face and Neck Laxity Grading Scale (FLR) [8], patient satisfaction questionnaires (PSQ) [12], the validated Fitzpatrick Wrinkle, Fold and Tissue Laxity Scale (FWFTLS) [18] and the Physician Global Assessment Scale (PHh-GAS) [20]. The effects of MFU treatment measured by these scales are comparable, i.e., a mild-to-moderate effect on tissue laxity.

### 3.5. Devices, Treatment Regimens and Adverse Effects (Table 2)

Eleven studies used the Ulthera system (Ulthera, Mesa, Arizona, USA) [8,12,13,14,17,18,21,24] and two used the Ultraformer system (Classys Inc., Seoul, Republic of Korea) [9,22]. The UTIMS A1 (Korust Co., Ltd., Seoul, South Korea) [16], the Doublo IFUS (Hironic, Yongin-si, Republic of Korea) [20] and the Microson (Microson, Cosmoplus Co., Sungnam, Republic of Korea) [19] devices were each evaluated once. An MFU treatment regimen consists of the systematic treatment of an area of skin area along different parallel treatment lines so that the multiple ‘shots’ are evenly distributed along these treatment lines in a specific area. The treatment can be performed at different tissue depths, typically 1.5 mm (dermal), 3 mm (subdermal) and 4.5 mm (the SMAS layer). The frequency of the ultrasound transducers ranges from 2 to 10 MHz, but the 4 and 7 MHz probes are used the most. The higher the MHz, the shorter the wavelength and the less deep the skin can be penetrated. The energy that accumulates in one spot (TIZ) ranges from 0.25 Joules (J) to 1.2 Joules, with the majority of settings ranging between 0.3–0.9 J. Multiple passes can be performed at different depths and directions. The total delivered energies can go up to 7200 J for full face treatment and 4600 J for the neck [18]. A total of 565 patients from 13 studies reported pain in all areas and at all device settings. Pain is rated on a 0–10 visual analog scale (VAS) or a numeric rating scale (NRS). The average reported pain score was in the 3.8 [2.5–6.1] range. The periorbital region and submandibular region treatments were more painful with higher VAS scores [10,18]. It should be noted that the pain was scored after applying topical anaesthetic ointment and/or systemic analgesics. Other side effects than pain were evaluated in a total of 573 patients. Transient erythema with or without oedema occurred in almost all the patients [13,14,18,22,23]. More uncommon were ecchymosis/bruising in four cases [9,11], transient dysesthesia in four cases [18,19], a wheal on the cheek in one case [10], skin burns in two cases [14], white linear striae of the neck in two cases [17] and one case of dermal white papules on the neck [24]. These uncommon adverse effects were noted in 2% of the total number of treated patients (14/573).

## 4. Discussion

This systematic review was undertaken to assess the effect of a single treatment of MFU on skin tightening. The overall results based on the IGAIS/SGAIS scores show that HIFU improved tightness of the skin to various degrees in most (>90%) of the included patients.

Clinically, the effect of a single MFU treatment results in long-term skin improvement. This can also be observed in the pooled IGIAS data (Table 5) with the percentage of patients in the ‘moderate improvement’ group increasing from 36% to 52% and the ‘mild improvement’ group decreasing from 47% to 34% during the 90 to 180 follow-up days. Support for this improvement is also seen in the pooled SGAIS data where the ‘no change’ category decreases from 25% at the 3-month, to 17% at the 6-month and 5% at the one-year follow-ups. MFU treatment entails applying numerous subdermal small heating points with temperatures of about 60–70 °C [24], resulting in localised denaturation of the collagen [25]. The denatured collagen proteins are then gradually replaced by newly formed collagen fibres; this neocollagenesis leads to thicker and tighter skin [4,6,26].

The prolonged effect of an MFU treatment may also be due to the remodelling process which begins on day 28, as demonstrated by Hantash et al. [6] and Keagle et al. [27]. The expression of HSP47, a heat shock protein involved in wound healing through fibroblast proliferation and collagen production via a STAT3 signalling pathway blockade, becomes elevated 3 months after the heat treatment [5,27]. This expression of the HSP47 heat shock protein implies neocollagenesis and may reflect the reason for the improvement in skin laxity/tightening persisting up to at least three months post-treatment. No particular area of the face seems to be most susceptible to MFU, though Sasaki et al. [18] noticed a higher response in the brow region compared to the nasolabial region when measured objectively. One study noted the highest response in the cheek area [16], whereas another study did not see great differences between the regions when assessed separately [11]. It seems quite difficult to assess separate areas of the face since the facial areas are connected to each other and each area has a different muscle tone and the subcutaneous tissue and skin vary in thickness. These may all influence the outcomes.

Multiple variables can be altered during MFU therapy. Different transducers can be used with different frequency and energy settings. The wavelength (MHz) determines the amount of tissue penetration. A higher frequency ultrasound penetrates tissue less deeply. The energy (Joules) applied determines the amount of tissue heating and is responsible for the effect. The number of treatment lines and TIZs can be altered, as can the direction of the treatment lines (linear horizontal, vertical, criss-cross). It seems that more energy and more treatment lines at different depths [18] increase the effectiveness. An in-depth discussion of the different settings and their possible effects are beyond the scope of this review.

Since this review only evaluated the results of a single MFU treatment, it would be interesting to know if multiple MFU treatments with different time intervals may have an additional effect on skin tightening. In theory, additional TIZ could be chosen during a second procedure, in the untreated dermis or in the SMAS for additional skin tightening. However, the problem of subjective bias is evident when assessing aesthetic results, whereas objective measurements are limited, and, when a certain ‘lift’ is measured, it does not mean that this is visible to the human eye or that it fits a more aesthetically pleasing outcome. To overcome a subjective bias, various studies chose to assess the IGIAS with a blinded approach whereby the reviewers had to first identify the pre- and post-treatment photographs and, when the correct post-treatment photo was identified, an ‘improvement’ grade could be given [10,13,15,17,19,23]. Alam et al. [17] asked two or three reviewers to evaluate the photographs and then only rated a result as ‘improved’ if there was an agreement between the two reviewers. It would be interesting to see if an alternative to this subjective scoring system could be developed, perhaps with an aesthetic score using 3D image evaluations or the use of artificial intelligence.

Although MFU seems to be highly efficacious for skin tightening, there are limitations to these results.

First, the presence of excessive skin laxity was an explicitly mentioned exclusion criterion [8,10,12,20]. Three studies measured skin laxity severity and demonstrated that improvement declined with increased baseline skin [10,18,20]. This suggests that MFU treatment is not so effective in more severe skin laxity cases, and a surgical approach would be the best option for them. Secondly, the included patients were mostly adult females (>90%), and it would be interesting to know if the results can be reproduced in men since male skin is different from female [28,29]. Thirdly, the presence of excessive subcutaneous fat [10,12] or a high BMI > 30 [12] was often taken as exclusion criteria. Two studies [10,24] concluded that *lower* BMI values positively influence the outcomes. Fabi et al. [24] reported a significantly higher SGAIS for patients with a BMI of ≤25 kg/m^2^. Oni et al. [10] reported less improvement in subjects with a BMI exceeding 30 kg/m^2^. In a later study, Werschler et al. [12] excluded all patients with a BMI exceeding 30 kg/m^2^, enforcing the implicated negative influence of a high BMI on treatment outcomes. A reason why a higher BMI leads to less therapeutic effect may be that the skin tension of such cases may be higher due to more facial volume. This higher tension would then counteract the skin shrinkage following MFU.

In conclusion, MFU treatment seems to be effective for tightening the skin of patients with mild-to-moderate skin laxity. It may be less for those with a BMI > 30. This could be corroborated by future studies that should also focus on male patients, on optimising treatment regimens and on ways to score treatment outcomes more objectively.

## Figures and Tables

**Table 1 ijerph-20-01522-t001:** Overview of studies included.

Study	Title	Type	Aim	N Age (Mean) Range	Whole Face/(Separate) Areas on the Face Evaluation	Method of Measurement	Length of Follow-Up
Alhaddad et al. [8]	Randomized, Split-Face, Evaluator-Blind Clinical Trial Comparing Monopolar Radiofrequency Versus Microfocused Ultrasound With Visualization for Lifting and Tightening of the Face and Upper Neck	Prospective, single-centre, randomized, evaluator-blinded, split-face clinical trial	To compare the efficacy and safety of MRF versus MFU-V for the lifting and tightening of the face and neck.	N = 20 100% female 52.6 32–60	Eyelids Cheeks Melolabial folds Jowls	Fasil Face and Neck Laxity Grading Scale (FLR) (clinician evaluation) 5-point Subject Global Aesthetic Improvement Scale (SGAIS) Pain Visual Analogue Score (VAS)	180 days
Ko et al. [9]	Efficacy and safety of non-invasive body tightening with high-intensity focused ultrasound (HIFU)	Prospective clinical trial	Evaluation of the efficacy and safety of HIFU for skin tightening on the face and body.	N = 32 91% female 44.4 21–59	Cheeks	Independent blinded evaluation by 3 reviewers SGAIS	12 weeks
Oni et al. [10]	Evaluation of a Microfocused Ultrasound System for Improving Skin Laxity and Tightening in the Lower Face	Prospective nonrandomized clinical trial	The authors investigated tightening and lifting of cheek tissue, improvement in jawline definition and reduction in submental skin laxity in patients treated with the Ulthera System.	N = 103 85% female 49.2 35–60	Cheek/lower face	Masked reviewers Patient satisfaction questionnaire Quantitative evaluation	90 days
Park et al. [11]	High-Intensity Focused Ultrasound for the Treatment of Wrinkles and Skin Laxity in Seven Different Facial Areas	Prospective	This study was aimed at evaluating the clinical efficacy of and patient satisfaction with HIFU treatment for wrinkles and laxity in seven different areas of the face in Asian skin.	N = 20 90% female 52.3 37–75	Whole face Supraorbital Zygomatic infraorbital Perioral Cheek Preauricular Jawline	Evaluation of pretreatment and post-treatment photographs by two independent clinicians. To assess the severity of facial wrinkles and skin measured by modified eight-point photographic scale. Each facial area was evaluated before treatment and after 3 and 6 months by using the following scale: 0, none; 1, mild; 2, mild/moderate; 3, moderate; and 4, severe. The overall clinical improvement was also assessed. Patient satisfaction score	6 months
Werschler et al. [12]	Long-term Efficacy of Micro-focused Ultrasound with Visualization for Lifting and Tightening Lax Facial and Neck Skin Using a Customized Vectoring Treatment Method	Prospective, open-label pilot study	To evaluate the efficacy and safety of patient-specific, customized micro-focused ultrasound with visualization treatment with vertical vectoring to lift and tighten facial and neck tissue.	N = 20 96% female 47 34–60	Whole face	Blinded qualitative assessment IGAIS, SGAIS Patient satisfaction questionnaires (PSQ) Quantitative evaluation	1 year
Lee et al. [13]	Multiple Pass Ultrasound Tightening of Skin Laxity of the Lower Face and Neck	Prospective study	To evaluate the efficacy and safety of patient-specific, customized micro-focused ultrasound with visualization treatment with vertical vectoring to lift and tighten facial and neck tissue.	N = 12 100% female 59 55–71	Lower face	Blinded reviewers evaluated paired pretreatment and post-treatment photographs IGAIS	90 days
Lu et al. [14]	Quantitative Analysis of Face and Neck Skin Tightening by Microfocused Ultrasound With Visualization in Asians	Single-site prospective, nonrandomized clinical trial	To evaluate the 800 treatment lines of MFU-V on skin tightening effect of face and neck in Asians using 2 quantitative analysis systems at 0, 90, and 180 days after treatment.	N = 25 92% female 53.3 40–61	Whole face	IGAIS blinded, SGAIS (live assessment of the subject with pretreatment digital image) Quantitative evaluation	180 days
Fabi et al. [15]	Retrospective Evaluation of Micro-focused Ultrasound for Lifting and Tightening the Face and Neck	Retrospective study	To evaluate the safety and efficacy of MFU with visualization (MFU-V) for noninvasive treatment of facial and neck skin laxity 180 days after treatment and determine what lifestyle factors affect treatment outcomes.	N = 48 100% Female 58 39–85	Whole face	Blinded reviewers IGAIS, SGAIS Patient satisfaction questionnaires (PSQ) Quantitative evaluation	180 days
Saket et al. [16]	Study of efficacy of esthetic High-Intensity Focused Ultrasound system on Iranian skin for reducing the laxity and wrinkles of aging	Not stated	To evaluate the clinical efficacy and safety of high-intensity focused ultrasound on skin laxity and wrinkles.	N = 22 100% female 35–62	Whole face Forehead Brow Infraorbital Nasolabial Perioral Lateral orbit Cheeks	The level of efficacy was evaluated and measured by observation of two reviewers from 10% to 100%, where 10% means no efficacy and 100 means maximum efficacy. Overall and regional measurements. Patient opinion	3 months
Alam et al. [17]	Ultrasound tightening of facial and neck skin: A rater-blinded prospective cohort study	Rater-blinded, prospective cohort study	To assess the efficacy of ultrasound skin tightening for brow-lifts in the context of a procedure treating the full face and neck.	N = 35 97% female 44 32–62	Brows	Three masked clinicians evaluated paired pre-treatment and post-treatment photos Quantitative evaluation	90 days
Sasaki et al. [18] Gr. 1	Clinical Efficacy and Safety of Focused-Image Ultrasonography: A 2-Year Experience	Prospective 2-part study	To assess the efficacy of ultrasound skin tightening for brow-lifts in the context of a procedure treating the full face and neck.	N = 107 94% female 53.5 25–77	Whole face Brows, nasolabial fold	Investigator Global Aesthetic Improvement Scale (IGAIS), blinded Quantitative evaluation The validated Fitzpatrick Wrinkle, Fold, and Tissue Laxity Scale (FWFTLS)	90 days
Sasaki et al. [18] Gr. 2				N = 55 96% female 64.4 26–74			90 days
Yalici-Armagan et al. [19]	Evaluation of microfocused ultrasound for improving skin laxity in the lower face: A retrospective study	Retrospective study	To evaluate the efficacy and safety of a newer microfocused ultrasound (MFU) device on the lower face laxity.	24 96% female 52.5 34–69	Lower face	Two blinded dermatologists independently assessed paired before and after photographs in a randomised fashion. IGAIS, SGAIS	median of 4.3 months
Friedman [20]	Intense focused ultrasound for neck and lower face skin tightening a prospective study	Prospective, single-center study	To report authors experience with Doublo IFUS (Doublo™, HIRONIC Co.,Gyeonggi-do, Korea) for treating neck and lower face laxity.	N = 43 91% female 56.5 24–80	Lower face	Physician global assessment scale (Ph-GAS) Physician global assessment scale (PHh-GAS): 0—worse, 1—0–25% poor response, 2—25–50% fair response, 3—50–75% good response, 4—75–100%–excellent response. Patient global assessment scale: 1—0–25% poor response, 2—25–50% fair response, 3—50–75% good response, 4—75–100%–excellent response Patient satisfaction: 0—not satisfied, 1—mildly satisfied, 2—moderately satisfied, 3—very satisfied.	90 days
Araco [21]	Prospective Study on Clinical Efficacy and Safety of a Single Session of Microfocused Ultrasound With Visualization for Collagen Regeneration	Prospective study	The primary study endpoint was the improvement of the laxity and ptosis face skin.	N = 50 94% Female 52.8 31–64	Mid/lower face	Reviewers scored the photographs from 1 to 20 by self-developed scoring system. Self-developed patient satisfaction questionnaire (PSQ)	6 months
Wanitphakdeedecha et al. [22]	The efficacy of macro-focused ultrasound in the treatment of upper facial laxity: A pilot study	Prospective, evaluator-blinded pilot study	To evaluate the efficacy and safety of MFU with a 2.0 mm transducer in the treatment of upper facial laxity in Thai patients.	N = 34 85% female 35.4 20–49	Upper face	Assessment of upper facial laxity improvement using a grading scale: 0 = no improvement, 1 = minimal improvement, 2 = moderate improvement, 3 = marked improvement, 4 = excellent improvement Quantitative evaluation	6 months
Shome et al. [23]	Use of Micro-focused Ultrasound for Skin Tightening of Mid and Lower Face	Prospective, double-blind study		N = 50 52% female 38.4 25–55	Mid/lower face	IGAIS, SGAIS	1 year

**Table 2 ijerph-20-01522-t002:** Overview of treatment protocols (device, settings, anaesthetic used), pain, and adverse effects.

Study/n	Intervention/Transducers	Settings/Lines/Joules (If Stated)	Anesthetics	Pain	Adverse Effects	Device Used
Alhaddad et al. [8] N = 20	Target the superficial musculoaponeurotic system of the face 4 MHz, 4.5 mm 7 MHz, 4.5 mm 7 MHz, 3.0 mm 10 MHz, 1.5 mm	A total of 195 lines were delivered to the deeper tissue level, and 205 lines were delivered to the superficial tissue level (one side of the face only)	Topical 7%/7% lidocaine–tetracaine topical ointment	2.35 ± 2.0 VAS (0–10)	One patient developed Grade 1 erythema.	Ulthera
Ko et al. [9] N = 32	n = 32. The sizes of the involved areas were 5.0 × 5.0 cm^2^ on each cheek MF1: 7 MHz, 1.5-mm MF3: 2 MHz, 3.0-mm MF4: 2 MHz, 4.5-mm	120 shots for the cheek, pulse ranged from 1.0 to 1.5 J distributing a total 537.6 J	Topical anaesthetic cream	3.00 ± 1.6 VAS (0–10)	Erythema was seen in up to 9.38% mostly subsided within 5 days. Ecchymosis was seen in up to 6.25% (n = 2) dissolving in 3 days.	Ultraformer III
Oni et al. [10] N = 103	All treatment areas received 2 passes: 1. Ulthera Deep See 4–4.5 transducer (deeper penetration) 2. DS 7–3.0 transducer (more superficial penetration) for the second pass	Approximately 295 exposure lines were placed on each patient’s face and neck.	Oral medications (5–10 mg of diazepam and 5/325 mg of hydrocodone/acetaminophen. Intramuscular medication (60 mg of ketorolac tromethamine)	Cheeks 5.68 Submental area 6.09 Submandibular region 6.53 NRS (0–10)	Wheal on cheeks in three patients.	Ulthera
Park et al. [11] N = 20	Patients were treated with a HIFU-tightening device to the entire face except for the nose and eyes. 4 MHz, 4.5-mm 7 MHz, 4.5-mm 7 MHz, 3.0-mm	Each probe delivered a set of pulses in a linear array at 1 cm intervals. From 400 to 500 shots were delivered according to the size of the face.	Topical lidocaine/prilocaine cream three patients, received a nerve block of the supraorbital, supratrochlear, intraorbital and mental nerves	Not stated	Six patients with erythema and swelling, and two patients with purpura and bruising. Resolved within 2 weeks.	Ulthera
Werschler et al. [12] N = 20	Treatments were delivered using a vectored pattern 4.0 and 7.0 MHz at focal depth 3.0 and 4.5 mm.	Subjects received a mean of 683 treatment lines (range 609–700) in the cheeks, submentum, submandibular, peri orbital and brow regions	Not stated	4.0 at 4.0 Mhz/4.5 mm 3.2 at 7.0 Mhz/3.0 mm, 5.5 at 7.0 Mhz/4.5 mm NRS (0–10)	One patient swelling under right eye. Resolved within 4 days.	Ulthera
Lee et al. [13] N = 12	The dermis and subcutaneous tissue were targeted using the 4-MHz, 4.5-mm-focal-depth and 7 MHz, 3.0 mm focal depth probes.	4 MHz, 4.5 mm focal depth (0.75–1.2 J) 7 MHz, 4.5 mm focal depth (0.75–1.05 J) 7 MHz, 3.0 mm focal depth (0.4–0.63 J	Topical anaesthetic ointment (9% lidocaine)	3.9 ± 1.66 VAS (0–10)	All subjects developed slight erythema and oedema immediately after treatment.	Ulthera
Lu et al. [14] N = 25	Subjects were treated with MFU-V to the face and neck using 2 different transducers: 4 MHz, 4.5 mm focal depth and 7 MHz, 3.0 mm focal depth with a total of 800 lines.	Total 800 lines were given; 4 MHz, 4.5 mm, 0.90 J, 350 lines on the cheeks and neck; 7 MHz, 3.0 mm 0.30 J 430 lines on the forehead temple area, cheeks and neck 7 MHz, 3.0 mm focal depth, 0.30 J, 20 lines on the infraorbital area.	All subjects had topical anaesthesia containing 2.5% lidocaine and 2.5% prilocaine oral analgesics with ibuprofen 800 mg before the treatment	4.1 (2.0) 4.5 mm 2.7 (1.6) 3.0 mm VAS (0–10)	Three soreness, 20 bruising/oedema/erythema, two others (contact dermatitis and submandibular burns).	Ulthera
Fabi et al. [24] N = 48	MFU-V treatment of the face and upper neck using the 4 MHz, 4.5 mm and 7 MHz, 3.0 mm depth transducers.	370–420 treatment lines at the highest energy settings.	10% of subjects received topical application of 23% lidocaine/7% tetracaine 15% received it in combination with oral diazepam (5–10 mg) The majority of subjects received a combination of topical anaesthesia, oral diazepam (5–10 mg) and an intramuscular injection of 50 to 100 mg of meperidine and 50 mg of hydroxyzine	Not stated	One patient showed evidence of a 2 mm white dermal papule on the upper neck.	Ulthera
Saket et al. [16] n = 22	Treatment of brow, forehead, infraorbital rim, lateral orbit, nasolabial folds, prioral and cheeks. The areas with the thinnest skin treated with superficial depth probes; the brow and temple treated with superficial and deeper probes; cheek and submental skin were treated with the deepest 4 MHz 4.5 mm probe followed by additional treatment with a superficial probe.	The energy level set between; 1.5 mm transducer 0.2 and 0.25 J; 3 mm transducer set between 0.5 and 0.7 J; 4.5 mm transducer set between 0.6 and 0.85 J. The number of shots (varied between 600 and 800 lines that seemed covered the whole faces with maximum efficacy.	None used	2.5 NRS 1–5	Not stated	UTIMS A1
Alam et al. [17] N = 35	Subjects treated with a focused intense ultrasound tightening device to the forehead, temples, cheeks, submental region and side of neck using the following probes: 4 MHz, 4.5 mm focal depth; 7 MHz, 4.5 mm focal depth and 7 MHz, 3.0 mm focal depth.	On average, 110 exposure lines were placed using the focused ultrasound system on the face and neck of each subject.	Topical anaesthetic Ointment (7%/7%) lidocaine-tetracaine	3–4 NRS (1–10)	Two early subjects developed elevated white linear striations of the neck.	Ulthera
Sasaki et al. [18] Gr. 1 N = 107	Above the superolateral brow, the fibromuscular layer and dermal treatment lines were administered in vertical directions, but these were administered horizontally within crow’s feet sites. Within the malar bag site, all fibromuscular and dermal treatment lines were placed in a superomedial direction. In the face and neck, fibromuscular treatment lines were positioned in a horizontal direction, and dermal treatment lines were placed superolaterally.	423 J to each lateral brow and crow’s feet (7 MHz, 3.0 mm, 15 lines; 7 MHz, 4.5 mm, 15 lines) 461.2 J to each malar bag (7 MHz, 3.0 mm, 15 lines; 4 MHz, 4.5 mm, 15 lines); 1845 J to each half of the face (7 MHz, 3.0 mm, 60 lines; 4 MHz, 4.5 mm, 60 lines); 2306 J to the entire neck (7 MHz, 3.0 mm, 75 lines; 4 MHz, 4.5 mm, 75 lines).	A pain management program was initiated in a graded fashion. It consisted of administering oral analgesic or sedative medication, giving distractive hand and foot massages, reducing skin temperature with an air coolant device, lowering joule settings (by 1 level for each transducer or by shortening the length of treatment lines) and, if necessary, administering selective nerve blocks or limited amounts of buffered lidocaine (subcutaneously)	Peri-orbital 5.7 face 3.7 NRS (0–10)	All patients experienced transient erythema for 1 to 2 h and mild swelling for several days. Mild bruising generally resolved within 1 to 2 weeks. Three patients had transient dysesthesia (numbness or hypersensitivity).	Ulthera
Sasaki et al. [18] Gr. 2 N = 55	Patients received twice the number of treatment lines (oppose to gr 1) and, therefore, increased joule energy to each site (except the malar bag area where treatment remained the same as before).	846 J to each lateral brow and crow’s feet (7 MHz, 3.0 mm, 30 lines; 7 MHz, 4.5 mm, 30 lines); 461.2 J to each malar bag (7 MHz, 3.0 mm, 15 lines; 4 MHz, 4.5 mm, 15 lines) 3690 J to each half of the face (7 MHz, 3.0 mm, 120 lines; 4 MHz, 4.5 mm, 120 lines); (3) 4612 J to the entire neck (7 MHz, 3.0 mm, 150 lines; 4 MHz, 4.5 mm, 150 lines).			Not stated	Ulthera
Yalici-Armagan et al. [19] N = 24	7.5 MHz 3.0 mm and 4 MHz 4.5 mm for treating lower facial and submental laxity. Treatment was performed by 2 dermatologists following the manufacturer’s recommended protocol.	Cheeks and submentum: 4.5 mm; 0.9 to 1.2 J and 3.0 mm 0.35 to 0.45 J. Mean treatment line 262 ± 29.7 (range 217–335).		No numeric measurement	One subject reported transient stinging sensation/dysesthesia on the face after the procedure that lasted approximately 6 months. Another subject reported erythema and striation after application.	Microson (Cosmoplus Co., Sungnam, Korea)
Friedman [20] N = 43	IFUS treatment of neck and lower facial skin laxity	The submental region, cheeks 4 MHz, 4.5 mm probe (1.2 J) and 7 MHz and 3.0 mm probe (0.65 J)	Topical anaesthetic ointment (lidocaine 2.5% and prilocaine 2.5%) oral 1 g acetaminophen	No numeric measurement 4 MHz, 4.5 mm probe was painful at times	Erythema and oedema were acute and transient responses. No numbers mentioned.	Doublo IFUS (Doublo™, HIRONIC Co.)
Araco [21] N = 50	The lower lids, zygomas, cheeks, submental area and mandibular lines were treated. 4.5 mm superficial muscular 3.0 mm aponeurotic system 1.5 mm subcutaneous tissue deep dermis	4 MHz (0.9 J) 4.5-mm 7 MHz (0.3 J) 3.0-mm 10 MHz (0.25 J) 1.5-mm All patients received 1200 spot lines 400 lines from each transducer at recommended energy power	Lormetazepam 2 mg, tramadol 25 mg and local lidocaine cream	3.32 ± 1.15 PPSR (10-point scale not validated)	Not stated	Ulthera
Wanitphakdeedecha et al. [22] N = 34	Patients were treated with a single session of MFU with 2.0 mm (5.5 MHz) transducer at the forehead, lateral and just below the eye area.	Total of 140 lines at 0.2–0.4 J; Forehead 90 horizontal lateral eye area five horizontal and vertical under eye area15 horizontal	Topical anaesthetic cream	3.03 ± 1.57 VAS (0–10)	All patients developed mild erythema immediately after the treatment with spontaneously resolved at 1-week follow-up.	Ultraformer III
Shome et al. [23] N = 50	Patients were treated 3.0 mm for deep dermis 4.5 mm for superficial muscular aponeurotic system.	7.5-MHz 3.0-mm forehead, 0.3 to 0.35 J; malar, 0.35 J; temple, 0.35 J. cheeks, submental areas; 4.4 MHz, 4.5 mm at 1.2 J; 7.5 MHz with the 3.0-mm 0.45 J. +/−500 exposure lines (range: 480–700)	Topical anaesthetic ointment (7%, lidocaine–prilocaine)	32% mild pain; 48% moderate pain, 20% severe pain (10-point scale (0 = no pain; 1–4 = mild pain; 5–8 = moderate pain; 9–10 = severe pain)	Almost all the patients had swelling that persisted for 2 to 14 days.	Ulthera

**Table 3 ijerph-20-01522-t003:** Quantitative analysis of MFU effects on facial skin tightening. Effects on brow lift, submental lift and marionette line lift.

		Quantitative Analysis				
	N=	Method of Measurement	Outcomes			
Oni et al. [10]	78	Submental lift. Fixed points were lateral canthus where the nostril meets the columella and where the chin meets the neck. For each lateral image, a line was first drawn horizontally from the lateral canthus (line a); a vertical line was then dropped down through the point where the columella meets the nostril (line b). An additional horizontal line was then drawn from line b to the point where the chin meets the neck; this line was then extended by 35 mm (line c). Finally, another vertical line was dropped from this point (line d). The area bounded by line c and line d, and the natural line of the neck (area x) was then calculated with AutoCAD software A reduction in area x represented tissue lift. A reduction of >20 mm^2^ denoted improvement.	At day 90, the average amount of lift was 45.2 mm^2^, reflecting improvement in skin laxity for 71.8% (56 of 78). Of the patients who experienced a quantitative lift, 82.1% (46 of 56) were Deemed improved according to the masked qualitative assessment, and 75.0% (42 of 56) noted improvement in their face and/or neck at day 90.			
Werschler et al. [12]	20	Brow/Submental lift. Quantitative assessments of brow and lower face tissue lift were completed using 2D photographs from all follow-up visits. Baseline and post-treatment photos were matched to ensure proper alignment. For the upper face, a lift measurement was considered improved if the eyebrow was raised ≥0.5 mm. For the lower face, an improved lift measurement was defined as a submental lift ≥1.0 mm. An improved measurement area was defined as a noticeably improved submental area ≥20 mm^2^ in size.	Day 90: 30 to 40 per cent reporting ≥1 mm lift on the right and left sides, respectively, decreasing to 22 to 33 per cent at one year. 40 to 50 per cent reported improvement over ≥20 mm^2^ at day 90 on the right and left sides, respectively, decreasing to 33 per cent for both sides at one year. The proportion of subjects with ≥0.5 mm eyebrow lift was 31 to 38 per cent on the right and left sides, respectively, 55 to 35 per cent at day 180 and 44 to 33 per cent at one year.			
Lu et al. [14]	25	Brow lift. The mean brow height was calculated as the average vertical distance of the medial canthus, medial limbus, lateral limbus and lateral canthus to the highest point of brow. The midcheek angle was the angle between a horizontal line drawn from alae nasi and a line drawn from alae nasi to the malar prominence. The calculations were made using the 3-dimensional imaging system. Submental lift was calculated according to Oni et al. [13]	There was a mean 0.47 mm brow lift at 90 days (*p* = 0.0165), but there was a 0.12 mm decrease in brow height compared to baseline at 180 days (*p* = 0.6494). At 90 days, a mean 26.44 mm^2^ submental lift was noted (*p* = 0.0217). In addition, at 180 days, a mean 13.76 mm^2^ submental lift was noted (*p* = 0.243).			
Alam et al. [17]	30	Brow lift. In the 0-degree views for each eye, 5 measurements of distance in millimetres were obtained from the line connecting both medial canthi to the top edge of the eyebrow by moving from the medial canthus laterally in 8 mm increments along the line horizontally bisecting the medial canthi. The maximum height and the average eyebrow height thus obtained were recorded.	The mean value of average change in eyebrow height as assessed by measurement of the photographs at 90 days was 1.7 mm, and the mean value of maximum change in eyebrow height was 1.9 mm			
Sasaki et al. [18] Pilot study 1	27	Brow lift/marionette lift. An average of three vertical displacements of each brow (midpupil, lateral canthus and lateral tail of brow) from the intercanthal horizontal axis or the average of three superolateral displacements of each marionette line along a fixed reference line (extending from inferior tragal notch to midpoint of marionette line) was used to compare measurements for each subject and between each group. Treatment of opposing brows and marionette folds by varying treatment protocols (vector directions and single/dual tissue treatment depths). Group 1–5 brows, group 6–9 marionette folds.	Group 1, 5.7 ± 1.2% vs. 1.0 ± 0.3%; Group 2, 6.6 ± 0.5% vs. 3.6 ± 0.7%; Group 3, 5.6 ± 1.3% vs. 2.4 ± 0.8%; Group 6, 3.8 ± 0.7% vs. 2.0 ± 0.5%; Group 7, 3.8 ± 0.7% vs. 1.8 ± 0.2% Group 4, 7.2 ± 1.4% vs. 3.7 ± 0.7%, Group 5, 6.0 ± 1.4% vs. 3.1 ± 0.9%, Group 7, 3.8 ± 0.7% vs. 1.8 ± 0.2%; Group 8, 2.4 ± 0.2%; vs. 1.1 ± 0.3%; Group 9, 2.7 ± 0.2% vs. 1.4 ± 0.2%			
Wanitphakdeedecha et al. [22]	27	Brow lift. The average eyebrow height was measured using ImageJ software by calculating the average vertical distance from the highest point of the eyebrow to the level of both midpupils in five positions per side (a; medial canthus, b; medial limbus, c; mid pupil, d; lateral limbus and e; lateral canthus to the highest point of the eyebrow)	The average mean difference in eyebrow height was significantly increased in all follow-ups when compared to the baseline (*p* = 0.000). The average eyebrow height elevation was 1.51 mm at 1-month, 1.25 at 3-month and 1.22 mm at 6-month follow-ups			
				Mean ± SD Mean Difference		*p-value*
				(cm)	(cm)	
			Baseline	2.95 ± 0.45		
			1 wk follow-up	3.05 ± 0.50	0.095 ± 0.015	0.000
			1 mo follow-up	3.10 ± 0.48	0.151 ± 0.016	0.000
			3 mo follow-up	3.08 ± 0.45	0.125 ± 0.016	0.000
			6 mo follow-up	3.07 ± 0.46	0.122 ± 0.017	0.000

## Data Availability

Data can be obtained through the corresponding author.

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
