# Peer review of "A Systematic Review of the Efficacy of Microfocused Ultrasound for Facial Skin Tightening"

_ijerph, 2023, doi:10.3390/ijerph20021522_

Round 1
Reviewer 1 Report
The authors report a systematic review on the effects of microfocused ultrasound for facial skin tightening. In conclusion, mild to moderate laxity of the skin can be treated by microfocused ultrasound. Thes treatment provides subjective and objective improvement after 90 days and after 1 year. The study is carefully done and the results clearly presented.
There are only few minor points that should be addressed.
Concerning the search strategy, the authors should clarify the Boolean relationship of the keywords (AND and OR, respectively (lines 58 - 63).
In the flow chart, the "carriage return" symbols and others have to be removed (Appendix 3).
In table 1, the column labeled "N" does not only provide "Age(mean)" but also range. This should be included in the heading.
Author Response
First of all, many thanks for taking your time for reviewing our manuscript.
Response to reviewer 1
There are only few minor points that should be addressed.
Concerning the search strategy, the authors should clarify the Boolean relationship of the keywords (AND and OR, respectively (lines 58 - 63).
>>> Additional text has been added in the Search Strategy as such:
‘The search was conducted to include the scope of ultrasound AND skin to select publications that deal with ultrasound and skin; to select publications that deal with skin rejuvenation more specifically different words addressing skin rejuvenation were selected so that at least one of them would be present (OR).’
In the flow chart, the "carriage return" symbols and others have to be removed (Appendix 3).
>>> We do not ‘see’ carriage return symbols in appendix 3, so unfortunately we could not address this. Hopefully the editorial office can address this.
In table 1, the column labeled "N" does not only provide "Age(mean)" but also range. This should be included in the heading.
>>> This is now included in the heading
Reviewer 2 Report
Thank you for giving me this opportunity to review this interesting study. This study tried to summarize the current evidence of the treatment effect of MFU therapy. I believe such a systematic review will be beneficial for many practitioners. The authors overall did a great job in searching papers and collecting information from them, but I have some concerns about how "treatment success" was defined. Please refer to my major comments below.
Major Comments:
1. The main concern of this study is how “the effect” of MFU therapy was determined. Although the qualitative evaluation is more practical, as you discussed, it can be biased if the evaluators were not blinded. Therefore, it might be important to reassess the results to see whether the improvement was still seen when non-blinded studies were excluded. The quantitative measurement should be less biased. However, I am not sure whether, for example, the absolute brow lift at 0.47 – 1.7 mm was enough for an aesthetically pleasing outcome.
2. I do not see any primary outcome measurement (= skin improvement) from some papers (Park, Saket, Friedman, Araco) in Table 3 – 5 and the main text. Please provide them.
3. Although it has been discussed that the setting of transducers (MHz/depth and Joules) can affect treatment outcomes based on the previous reports (Line 133-137), a such relationship was not evaluated in this study. If you have done the such analysis, please provide/discuss it.
Minor Comments:
Results
3.3 Study characteristics
1. The last sentence says, “Six studies evaluated the face as a whole whereas 12 studies analyzed parts of the face (also) separately." 6 + 12 = 18. The total number of analyzed papers was 16. Does this mean there was an overlap?
Table 1
2. It says "Not stated" for a type of study of Yalici-Argamen et al 2020 paper, but the title clearly says “A retrospective study.” Wasn’t it a retrospective study?
Table 4
3. What does “Any #” at the top of each column mean?
Appendix 3
4. The number of “Records – screened” was 693 and “Records-excluded” was 643. 693 – 643 = 40. Then, why was the number of “Full-text-articles-assessed-for-eligibility” 43?
5. Also, please provide the exclusion reasons for 643 “Records-excluded”? I guess they did not have “any of the eligibility criteria appeared in the title or abstract” based on the material and method, 2.3. Study selection section, but it would be nice to make it clear.
6. Why was the excursion criterion for 4 of “Full-test-articles-excluded, -with-reasons-“ follow-up <180-days or 3-month? The exclusion criterion for the follow-up period was >3 months, not >180 days, according to Appendix 2.
Author Response
Dear Madam, Sir, many thanks for your time and effort to review our manuscript.
Response to reviewer 2.
Major Comments:
- The main concern of this study is how “the effect” of MFU therapy was determined. Although the qualitative evaluation is more practical, as you discussed, it can be biased if the evaluators were not blinded. Therefore, it might be important to reassess the results to see whether the improvement was still seen when non-blinded studies were excluded. The quantitative measurement should be less biased. However, I am not sure whether, for example, the absolute brow lift at 0.47 – 1.7 mm was enough for an aesthetically pleasing outcome.
>>> Qualitative aspect: this refers to the IGAIS for which the results are presented in table 4. The studies included in this results are all blinded studies. This was not always mentioned in the summarizing tables, hence the confusion. This has now been corrected.
>>> Quantitative aspect: Concerning your question about brow lifting, small differences may well be noticed and may therefore be of importance. For example in eyelid surgery, a difference of 1 mm is considered clinically relevant.
- I do not see any primary outcome measurement (= skin improvement) from some papers (Park, Saket, Friedman, Araco) in Table 3 – 5 and the main text. Please provide them.
>>> The primary outcome measurements are now provided in the overview table for completeness, so they have not to be repeated elsewhere.
- Although it has been discussed that the setting of transducers (MHz/depth and Joules) can affect treatment outcomes based on the previous reports (Line 133-137), a such relationship was not evaluated in this study. If you have done the such analysis, please provide/discuss it.
>>>> No relationship was discussed/provided since this was not the purpose of this review. This is now briefly addressed in the tekst.
Minor Comments:
Results
3.3 Study characteristics
- The last sentence says, “Six studies evaluated the face as a whole whereas 12 studies analyzed parts of the face (also) separately.” 6 + 12 = 18. The total number of analyzed papers was 16. Does this mean there was an overlap?
>>> Yes, this is an overlap
Table 1
- It says "Not stated" for a type of study of Yalici-Argamen et al 2020 paper, but the title clearly says “A retrospective study.” Wasn’t it a retrospective study?
>>> This was indeed a retrospective study, corrected as such
Table 4
- What does “Any #” at the top of each column mean?
>>> This means “Any improvenment”. We realize that this may be confusing so we left it out of the tables.
Appendix 3
- The number of “Records – screened” was 693 and “Records-excluded” was 643. 693 – 643 = 40. Then, why was the number of “Full-text-articles-assessed-for-eligibility” 43?
>>> We made a typing error: 643 should be 640; this is corrected
- Also, please provide the exclusion reasons for 643 “Records-excluded”? I guess they did not have “any of the eligibility criteria appeared in the title or abstract” based on the material and method, 2.3. Study selection section, but it would be nice to make it clear.
>>> indeed, this is now made clear
- Why was the excursion criterion for 4 of “Full-test-articles-excluded, -with-reasons-“ follow-up <180-days or 3-month? The exclusion criterion for the follow-up period was >3 months, not >180 days, according to Appendix 2.
>>> studies with less than 3 months follow-up (less than 90 days) were excluded, this was indeed inconsistent, it has now been corrected.
Reviewer 3 Report
This article is a systematic review of the efficacy of micro-focusedultrasound (MFU) for facial skin tightening. They did a thorough search of theliterature and their data are well presented.The main question of this article is to systematically review theefficacy of micro-focused ultrasound (MFU) for facial skin tightening. They systematicallysearched Pubmed, and Embase to assess the efficacy of single MFU treatments for facialskin tightening. To date, there is no systematic review of this topic.
The paper is well written, and the reader can easily understand whatis written. Theconclusions are consistent with the evidence and arguments presented. Theyconclude that MFU treatmentseems to be effective for tightening the skin of patients with mild to moderateskin laxity. It may be less for those with a BMI > 30.
Author Response
Thank you very much for your kind review.
Round 2
Reviewer 2 Report
Thank you for responding to my comments. They now became much clearer. I saw you added more information in Table 1 to respond to my major comment #2, but I still do not see the “results” of primary outcome measurement for papers by Park, Saket, Friedman and Araco whose results were not in Table 3-5.
Author Response
Response to reviewer 2.
Thank you for responding to my comments. They now became much clearer. I saw you added more information in Table 1 to respond to my major comment #2, but I still do not see the “results” of primary outcome measurement for papers by Park, Saket, Friedman and Araco whose results were not in Table 3-5.
>>> Dear reviewer, thank you for your reply. The text below is added in the main tekst. We deliberately keep the information compact, to prevent that too much attention is given to these studies.
Park et al. (2015) used a self modified photographic scale evaluating 7 area’s of the face in 20 patients. Each area was scored from 0 (no wrinkles) to 4 (severe). A HIFU treatment decreased the wrinkle score by 0.9 points at 3 and 6 months, with most effect on the jaw-line and cheek area.
Saket et al. (2017) evaluated MFU effect by an ‘observation’ scoring system ranging from 10 (no efficacy on wrinkles and lifting effect) to 100 (maximum efficacy). After 3 months post treatment, most of the 22 patients treated scored between 58 and 66 (physycian rated) or 50-60 (patient rated).
Friedman et al. (2020) treated lower face and neck laxity with ultrasound in 43 patients. Based on a global grading scale from 0 (exacerbation) to 5 (75%-100% improvement), at three months an improvement was seen in 9 patients that had only slight sagging. The others showed no effect.
Araco (2020) used his self developed Surgeon Assessment Scoring System and patient satisfaction questionnaires in 50 patients having one MFU treatment. After 6 months the surgeon score was 80 indicating a moderate effect on skin texture and a minimal to moderate effect on face lifting as visible on photographs. Patients indicated a ‘moderate’ to ‘good’ difference after treatment.